More of the same: allopatric humpback whale populations share acoustic repertoire

Fournet Michelle E.H. michelle.fournet@gmail.com 1 2
Jacobsen Lauren 3
Gabriele Christine M. 4
Mellinger David K. 2
Klinck Holger 3
1 Department of Fisheries and Wildlife, Oregon State University , Corvallis , OR , United States of America
2 Cooperative Institute of Marine Resource Studies, Oregon State University and NOAA Pacific Marine Environmental Laboratory , Newport , OR , United States of America
3 Bioacoustics Research Program, Cornell Lab of Ornithology, Cornell University , Ithaca , NY , United States of America
4 Humpback Whale Monitoring Program, Glacier Bay National Park and Preserve , Gustavus , AK , United States of America
Zimmermann Elke
Electronic publication date: 2018 Jul 30
Publication date: 2018
Volume: 6
Electronic Location ID: e5365
Received 2018 May 6; Accepted 2018 Jul 11
Copyright: ©2018 Fournet et al.
Copyright year: 2018
Copyright holder: Fournet et al.
License: This is an open access article distributed under the terms of the Creative Commons Attribution License, which permits unrestricted use, distribution, reproduction and adaptation in any medium and for any purpose provided that it is properly attributed. For attribution, the original author(s), title, publication source (PeerJ) and either DOI or URL of the article must be cited.
License URL: https://creativecommons.org/licenses/by/4.0/

Keywords: Humpback whale, Non-song vocalizations, Innate calls

Funding: National Parks Foundation Alaska Coastal Marine Grant program Hatfield Marine Science Center, Oregon Sea Grant Oregon Chapter of the Wildlife Society This work is funded by the National Parks Foundation Alaska Coastal Marine Grant program, the Hatfield Marine Science Center, Oregon Sea Grant, and the Oregon Chapter of the Wildlife Society. The funders had no role in study design, data collection and analysis, decision to publish, or preparation of the manuscript.

==============================
Background

Humpback whales (Megaptera novaeangliae) are a widespread, vocal baleen whale best known for producing song, a complex, repetitive, geographically distinct acoustic signal sung by males, predominantly in a breeding context. Humpback whales worldwide also produce non-song vocalizations (“calls”) throughout their migratory range, some of which are stable across generations.

Methods

We looked for evidence that temporally stable call types are shared by two allopatric humpback whale populations while on their northern hemisphere foraging grounds in order to test the hypothesis that some calls, in strong contrast to song, are innate within the humpback whale acoustic repertoire.

Results

Despite being geographically and genetically distinct populations, humpback whales in Southeast Alaska (North Pacific Ocean) share at least five call types with conspecifics in Massachusetts Bay (North Atlantic Ocean).

Discussion

This study is the first to identify call types shared by allopatric populations, and provides evidence that some call types may be innate.

Introduction

The study of acoustic signaling is a valuable tool for investigating animal behavior across a broad range of taxa (Brockelman & Schilling, 1984; Gannon, 2008; Pijanowski et al., 2011; Clink, Crofoot & Marshall, 2018). Sounds produced by animals can be systematically measured and compared, as can patterns of vocal behavior made in association with critical activities such as breeding, foraging, or socializing. Acoustic monitoring allows for broad-scale observations of animals across space and time and between populations (Mann & Lobel, 1998; Cerchio, Jacobsen & Norris, 2001; Risch et al., 2007; Potvin, Parris & Mulder, 2011). When coupled with what is known about genetics, population structure, and behavior, acoustic analyses become powerful tools for investigating the factors that shape communication signals.

Drivers of acoustic repertoires vary between taxa and species. While anatomy is a restricting force driving sound production, genetic, neurological, and environmental drivers also influence acoustic repertoires and vocal plasticity. For acoustic communication to be effective, a sound must be detectable within its acoustic habitat and sufficiently convey information to a receiver. As such, acoustic communicators have evolved adaptations to couple the acoustic properties of sounds to the environment in which they are produced in order to meet their signaling needs and maximize fitness (Slater, 1983; Boncoraglio & Saino, 2007). As a result, within the repertoire of most, if not all, sound-producing vertebrates are a collection of innate (i.e., unlearned) calls that are exercised independently of vocal learning and persist across generations (e.g., Domestic Fowl Gallus gallus and other species in the order Galliformes (Konishi, 1963; Matsunaga & Okanoya, 2009), white-handed gibbons Hylobates lar (Brockelman & Schilling, 1984), New Zealand fur seals Arctocephalus forsteri (Page, Goldsworthy & Hindell, 2001)). A smaller subset of taxa—most notably passerine songbirds—exhibit a combination of learned and unlearned vocal signals, which persist over time within a population (Baker & Jenkins, 1987; Vicario, 2004; Matsunaga & Okanoya, 2009; Zann, 2010). Some mammals including pinnipeds (taxonomic group including seals and sea lions), and cetacean species (taxonomic group including whales, dolphins, and porpoises) are also capable of vocal learning as indicated by vocal imitation or improvisation (Tyack & Sayigh, 1997; Poole et al., 2005; Petkov & Jarvis, 2012). What is less well described among mammalian vocal learners, however, is the coupling of stable sound types, which may be innate, with a dynamically changing repertoire of sound types whose variation appears to be culturally driven. Cetaceans, and specifically humpback whales (Megaptera novaeangliae), may be the best example of a taxon which exhibits this coupling of highly stable calls types and dynamically shifting vocal behaviors (Payne & Payne, 1985; Tyack & Sayigh, 1997; Rekdahl et al., 2013; Fournet et al., 2015; Fournet, 2018).

Humpback whales are a migratory baleen whale with a cosmopolitan distribution. Generally, humpback whales migrate between low-latitude breeding and calving grounds and high-latitude foraging grounds (Clapham, Mead & Gray, 1999). Their vocal behaviors are geographically and seasonally stratified. Primarily on breeding grounds and migratory corridors, but also to a lesser extent on foraging grounds, male humpback whales produce a long elaborate, and repetitive vocal display known as ‘song’, (Payne & McVay, 1971; Gabriele & Frankel, 2002; Stimpert et al., 2012; Dunlop & Noad, 2016; Herman, 2017). Songs are highly structured and acoustically complex, and are culturally transmitted between males within a single breeding region (Cerchio, Jacobsen & Norris, 2001; Mercado, Herman & Pack, 2005; Herman et al., 2013; Herman, 2017). Song structure changes rapidly over time (1–2 years) (Payne & Payne, 1985; Noad et al., 2000; Parsons, Wright & Gore, 2008). Further, geographic variation in song between regions is typical (Winn et al., 1981; Cerchio, Jacobsen & Norris, 2001; Parsons, Wright & Gore, 2008), with song sharing only occurring between regions that share individuals (Cerchio, Jacobsen & Norris, 2001; Mercado, Herman & Pack, 2005; Garland et al., 2015; Herman, 2017).

Humpback whales of both sexes and across the migratory range also produce a series of vocalizations (“calls”) independently of song (Silber, 1986; Dunlop, Cato & Noad, 2008; Stimpert et al., 2011). Calls occur in isolation or in short bouts and occasionally appear as song units (Rekdahl et al., 2013; Rekdahl et al., 2015). Call use varies based on social and behavioral context; some calls facilitate intra-group interactions, while other calls are specific to foraging contexts (Stimpert et al., 2007; Dunlop, Cato & Noad, 2008; Wild & Gabriele, 2014; Fournet et al., 2018). Unlike song, many calls are stable over time. The most commonly produced call types in the east Australian migratory corridor, making up 64% of the call detected in one study, are stable over 7–11 year time periods (Rekdahl et al., 2013), while in Southeast Alaska, at least 16 call types, including all described call types to date, persist in the call repertoire for decades and across generations (Fournet et al., 2015; Fournet, 2018).

Call longevity across generations is an indication that some call types may be fixed within the humpback whale repertoire. Identifying the same stable call types in other, unrelated populations would provide further evidence that humpback whales may be anatomically or behaviorally predisposed toward the production of certain sounds. Qualitative comparisons have been made of calls produced in the North Pacific (Southeast Alaska, USA), South Pacific (East Australia), North Atlantic (Massachusetts Bay, USA) and South Atlantic (Coastal Angola, Africa) with the general agreement that global humpback whale populations produce some similar call types (Dunlop et al., 2007; Stimpert et al., 2011; Fournet et al., 2015; Rekdahl et al., 2016), but no formal comparison of call types between populations has been thus far attempted.

To test the hypothesis that some calls types are inherent to humpback whales, we looked for evidence of shared call types in the call repertoire of two allopatric humpback whale populations on their northern latitude foraging grounds, one in the North Atlantic and one in the North Pacific. Based on genetic analyses it is estimated that global humpback whale populations last shared a maternal ancestor in the Miocene, approximately 5 Mya, and that discrete lineages split 2–3 Mya (Baker et al., 1993). In the northern hemisphere, humpback whales in the Atlantic and Pacific Ocean are geographically separated by the North American continent and are genetically isolated from one another (Valsecchi et al., 1997; McComb et al., 2003). Cultural exchange of acoustic signals between the two populations is extremely unlikely based on this geographic barrier and known migratory patterns. Thus, a shared acoustic repertoire would indicate that individual signals may be fixed within the species and conserved with time, rather than socially learned. We hypothesized that call types that are stable across multiple generations on a North Pacific foraging ground would also be present in the humpback whale call repertoire on a North Atlantic foraging ground.

Methods

Data collection

We compiled acoustic datasets from two humpback whale foraging grounds in the North Pacific and North Atlantic. Acoustic data from Southeast Alaska (SEAK: North Pacific) were collected using passive acoustic recording devices during summer months (June–August) in Frederick Sound in 1976 and Glacier Bay National Park and Preserve (GBNPP) in 2007, and 2008. Acoustic recordings were also collected using passive acoustic recording devices deployed during summer months in Massachusetts Bay (MB; North Atlantic) in 2008 (Fig. 1, Table 1). Acoustic recordings from Frederick Sound, SEAK were opportunistically collected with a dip hydrophone from a drifting vessel and were of variable duration (32–94 min). Acoustic recordings from GBNPP made in 2007 and 2008 were collected from a cabled hydrophone in Bartlett Cove (Fig. 1) with a 30-seconds-per-hour recording cycle (Wild & Gabriele, 2014). Data from GBNPP were reviewed by US Navy acousticians to characterize the content of each sound sample. Data from MB were collected as part of a long-term monitoring project in that region (see also Hatch et al., 2012). Recordings were made using an array of marine autonomous recording units (MARUs; Calupca, Fristrup & Clark, 2000; Table 1). Research analysts from the Bioacoustics Research Program at Cornell Laboratory of Ornithology reviewed array recordings and noted the presence or absence of humpback whale calls on each element. We randomly subset 60 h of two-channel acoustic data from the array for analysis (Fig. 1). Sound samples from both regions were analyzed only if they were known to contain humpback whale calls.

Figure 1 Map of (A) Southeast Alaska, North Pacific recording locations and (B) Massachusetts Bay, North Atlantic recording locations.

Red area indicates sampling region for hydrophone recordings made in 1976. Stars in both maps indicate moored hydrophone locations. Map data© 2016 Google.

Data processing and analysis

Recordings from SEAK were originally sampled at 44.1 kHz and were resampled at a rate of 2 kHz for consistency with data from MB (Table 1); all recordings were made with 16 bit resolution. Once resampled, recordings were comparable, though not completely equivalent. Differences in recording equipment and conditions may manifest in extracted feature values; however when paired with robust call inclusion criteria and our choice of feature extraction methodology (see below) call classification is robust to these differences. Spectrograms of acoustic recordings were created with Raven Pro 1.5 (Cornell Lab of Ornithology, Ithaca, NY) using an FFT length of 1,046, a 30 s window length, Hann window, 75% overlap, for a frequency resolution of 2.75 Hz and constrained to the 10 Hz–1 kHz frequency range to facilitate analysis. Recordings were manually reviewed by experienced observers familiar with the humpback whale call repertoire. All calls were annotated in the time-frequency domain and salient acoustic features were extracted for quantitative classification in Raven Pro (Table 1). Aggregate entropy was also extracted for each sound (Table 1). In some cases differences in aggregate entropy reflect variation in recording conditions; however, where considerable differences in acoustic structure exist (e.g., between call types) aggregate entropy is one of the few acoustic measurements capable of discriminating between structurally ‘simple’ calls (see droplets in Fig. 2B) and structurally ‘complex’ calls (see whups Fig. 2D). For this reason when recording conditions vary, aggregate entropy is still relevant for discriminating between-call type differences, which are generally more contrasting. In this study data exploration did not reveal any significant differences in aggregate entropy related to recording location or year.

Table 1 Recordings specifications for data collection protocols from North Pacific and North Atlantic foraging grounds.

Year	1976	2007 & 2008	2008	
Hydrophone model	Unknown	ITC 8215A	HTI-94-SSQ	
Sampling rate	44.1 kHz	44.1 kHz	2 kHz	
System sensitivity	Unavailable	−174 dB ± 2 dB re 1 V/μPa	−168 dB ± 1 dB re 1 V/μPa	
Deployment method	Dipping (20 m)	Bottom-mounted (52 m)	Bottom-mounted (∼60 m)	
Location	Frederick sound	Glacier Bay	Stellwagen Bank National Marine Sanctuary	
Recording cycle	Non-standardized	30 seconds from every hour	Continuous	
Data format	Continuous	30-second recordings	5-minute recordings	
Recording days	4	72	10	
Date range	July 1976	June–September 2007 June–September 2008	June–August 2008	

Figure 2 Spectrograms of call types by ocean basin (FFT 256, Hann window, 90% overlap).

Call types: (A) swops, (B) whups and growls, (C) teepees, (D) droplets. The horizontal lines at ∼500 and 800 Hz in spectrograms from the Atlantic indicate vessel noise.

For harmonic sounds, measurements of the start- and end-frequencies were made on the fundamental frequency. For amplitude-modulated sounds containing a broadband component, measurements were made on the lowest-frequency component of the call (Dunlop et al., 2007; Rekdahl et al., 2013). Frequency parameters were log-transformed to account for the mammalian perception of pitch (Table 2) (Richardson et al., 1995; Dunlop et al., 2007); (Fournet, Szabo & Mellinger, 2015); although humpback pitch perception has not been studied experimentally, humpback ear morphology suggests that their sound reception is, like other mammals, approximately logarithmic (Southall et al., 2007).

Table 2 Acoustic parameters used in Classification and Regression Tree (CART) analysis.

Duration (90%) (s)	90% of the duration of the annotated call	
Bout	Number of repetitions of the same call type	
Low frequency (Hz)*	Lowest frequency component of the call	
High frequency (Hz)*	Highest frequency component of the call	
Bandwidth (90%) (Hz)	90% of the difference in frequency between high and low frequency	
Start frequency (Hz)*	Starting frequency of fundamental	
End frequency (Hz)*	Ending frequency of fundamental	
Peak frequency (Hz)*	Frequency of the spectral peak	
Center frequency (Hz)*	The frequency that divides the sound equally into two intervals of equal energy	
Frequency trend*	Start F0/End F0	
Aggregate entropy (bits)	A measure of total disorder in the call (RavenPro, 1.5)	
Notes.

Log transformed parameters are indicated with an asterisk (*).

Time-frequency parameters were input into a Principal Component Analysis (PCA) in order to aggregate correlated variables for classification and comparative analyses (R, psych package). A varimax rotation was applied (Table 2) to maximize loading and facilitate variable interpretation (Cerchio & Dahlheim, 2001; Dunlop et al., 2007). By pairing PCA values with traditional acoustic measurements during classification analyses we account for the broad structure of the call (e.g., broadband and high frequency vs. narrowband and high frequency) as well as the fine-scale acoustic features. Boxplots (median, first, and third quartile PCA values) were generated in R using the ggplot2 package (Wickham, 2016) to qualitatively compare differences in call structure between ocean basins.

Signal-to-noise ratios (SNRs) were calculated for each acoustic sample by measuring the in-band power contained in a one-second sound sample directly preceding each call; this value was then subtracted from the in-band power measured of the call of interest to get the band-limited SNR value. Calls in this study were only included if they had a SNR of 10 dB or higher (Dunlop et al., 2007; Rekdahl et al., 2016).

Using the existing SEAK call catalog as a reference, each acoustic sample was assigned an a priori call type based on aural and visual call features. Because the goal of this study was to investigate the potential for call types to be fixed within this species, only call types that persist across generational timescales that could be detected given a 2 kHz sampling rate were included in this study; this included droplet, growl, swop, teepee, whup, and feeding calls (Fournet et al., 2015; Fournet, 2018). Droplets, swops, and teepees are short-duration pulsed calls that typically occur in short repeated sequences. Growls and whups, by contrast, are harmonic and amplitude-modulated calls that are generally not repeated (Wild & Gabriele, 2014; Fournet et al., 2015; Fournet, 2018). Feeding calls are stereotyped highly-tonal, low-complexity calls that have been closely associated with herring foraging in SEAK humpback whales (Cerchio & Dahlheim, 2001; Fournet et al., 2018). Acoustic samples that were qualitatively different than previously described call types were classified as ‘unknown’ and no further attempts for classification were made. Initial data exploration found no significant differences in acoustic parameters of calls recorded in GBNPP and calls recorded in Frederick Sound; calls from SEAK were pooled for analysis.

Quantitative classification methods were identical to those used by Fournet (2018), with the exception that all predictor variables were extracted in Raven Pro. For consistency with other humpback whale call classification studies, calls were classified through the use of a Classification and Regression Tree analysis (CART, rPart package; R Development Core Team, 2013) and a random forest analysis (randomForest package; Liaw & Weiner, 2007) using the methodologies described by Rekdahl et al. (2013) and Rekdahl et al. (2016). The combination of CART and random forest analyses to validate human call type assignment has emerged as the preferable method for classification of humpback calls (as well as other cetaceans; Garland, Castellote & Berchok, 2015). CART analyses are robust to outliers, non-normal and non-independent data, and random forest analyses improves accuracy using a bootstrapping method to generate a level of uncertainty for each classification tree, rather than a single classification tree (Breiman et al., 1984; Rekdahl et al., 2013; Rekdahl et al., 2016). In the CART analysis the Gini index was used to assess the “goodness-of-split” for each node in the tree. All variables were considered independently and ranked, and the splitting variable that minimized splitting error was selected (Breiman et al., 1984; Rekdahl et al., 2013; Rekdahl et al., 2016). Terminal nodes were set to have a minimum sample size of ten. Trees were overgrown and then pruned upward until reaching the tree with the lowest misclassification rate (Breiman et al., 1984). A total of 1,000 trees were grown for the random forest analysis. Predictor variables included salient acoustic features as well as two rotated principal components (PC) that aggregated correlated acoustic variables (Dunlop et al., 2007); a detailed description of predictor variables can be found in Table 2. Quantitative classification assignments were compared to a priori call type assignments to validate observer classification. Major discrepancies in call type assignment were re-reviewed by at least two observers. Calls were excluded if observers were not in agreement. If observers were in agreement about call type assignment than the a priori classification was deemed ‘correct’. All analyses were conducted in R version 3.3.3 (R Development Core Team, 2013).

To assess differences in acoustic parameters between calls from MB and SEAK populations, we summarized and compared PC values for all call types that exhibited stability between regions. Comparative analyses were made based on a priori classification. Humpback whales in both SEAK and MB exhibit seasonal movements throughout foraging grounds during summer months (Baker et al., 1985; Straley, Gabriele & Baker, 1995; Weinrich, 1998; Payne et al., 1986; Schilling et al., 1992), reducing the likelihood of their repeated acoustic capture on hydrophones, which have a finite listening range. Additionally, a random subset of acoustic data spanning summer months in MB was selected for analysis in order to reduce the likelihood of repeated capture of individuals. Also, the temporal breadth of recordings made in in SEAK (Table 1) make the probability of documenting only a small subset of individuals from this region unlikely. However, because data were collected passively without concomitant visual observations the number of vocalizing individuals is unknown. For this reason the independence of each data point cannot be confirmed and statistical tests pertaining to population level differences are inappropriate.

Results

A total of 411 sounds fitting the inclusion criteria were classified to one of six known call types (droplets, growls, feeding calls, swops, teepees, whups; Fournet, Szabo & Mellinger, 2015); 191 calls were collected across 10 recording days from Massachusetts Bay (MB), and 220 calls were collected across 76 sample days from Southeast Alaska (SEAK; Tables 1 and 3). Drops, growls, swops, teepees, and whups were found in both populations (Fig. 2, Table 3); feeding calls were detected only in SEAK. A Bartlett’s Test of Sphericity indicated that data was suitable for factorial analysis (χ2 = 18, 106.78, d.f. = 55, p < 0.00001); this was confirmed by a Kaiser-Meyer-Olkin value of 0.61. The first rotated component (PC1) corresponded most closely to aggregate entropy, bandwidth, and upper frequency (proportion variance explained = 0.51), meaning that as PC1 increases, the calls grow more complex, grow broader-band, and extend to higher frequencies. The second rotated component (PC2) corresponded most closely to lower frequency, start frequency, and peak frequency (proportion variance explained = 0.49), meaning that as PC2 increases, calls grow higher in pitch overall, but not necessarily more broadband or complex. Neither component was strongly affiliated with duration or bout in this analysis, meaning that the PC variables in this analysis do not represent temporal variability.

Table 3 Summary statistics (mean in bold, standard deviation) for call parameters by call type and location.

Type	Variable	Atlantic	Pacific	
Low frequency harmonic	Growl	N	41	78	
Low freq (Hz)	41.5	12.2	35.8	21.8	
Peak freq (Hz)	87.4	15.1	116	62.6	
Duration (s)	0.8	0.24	0.7	0.3	
Whup	N	21	36	
Low freq (Hz)	49.9	15.8	47.4	25.1	
Peak freq (Hz)	94.9	26.2	128	70.3	
Duration (s)	0.6	0.18	0.7	0.2	
Pulsed	Droplet	N	44	29	
Low freq (Hz)	99.4	49	148	99.8	
Peak freq (Hz)	187	62.6	252	120	
Duration (s)	0.4	0.2	0.3	0.16	
Swop	N	45	16	
Low freq (Hz)	76.5	31.4	70	30	
Peak freq (Hz)	159	54.3	214	85.6	
Duration (s)	3.9	4.2	0.3	0.2	
Teepee	N	40	51	
Low freq (Hz)	40	17	214	25.1	
Peak freq (Hz)	79.2	28.8	154	70.3	
Duration (s)	1.1	1.77	0.4	0.23	

CART call type assignment and a priori call type assignment were in agreement 82% of the time (n = 335∕411, Table 4). In descending order of importance, splitting variables for CART classification were bandwidth, bout, center frequency, duration, end frequency, aggregate entropy, lower frequency, and PC1. The random forest analysis correctly classified most of the calls (out-of-bag error rate = 23%). The variables most important for splitting decisions in the random forest analysis in were bout, end frequency, duration, aggregate entropy, lower frequency, PC1, PC2, and frequency trend, in descending order of importance. Whups were the most commonly misclassified calls (Table 4); in the CART analysis whups were mistaken for growls 38% of the time (n = 22). Observers validated call type assignment for most whup calls (95%, n = 57); three calls were omitted due to classification incongruity.

Table 4 Confusion matrix indicating agreement between (vertical) Classification and Regression Tree call type assignment versus (horizontal) human call type assignment.

	Droplet	Feed	Growl	Swops	Teepee	Whup	Agreement	
Droplet	58	0	3	5	4	3	79%	
Feed	0	10	0	0	0	0	100%	
Growl	0	0	111	1	3	4	93%	
Swops	5	0	1	44	9	2	72%	
Teepee	3	0	3	4	81	0	89%	
Whup	2	0	22	2	0	31	54%	
					Total agreement	82%	

PC1 values were higher in SEAK than MB for all call types except for growls, indicating that calls from SEAK were generally broader band and exhibited higher levels of complexity (Table 3, Fig. 3). PC2 values were higher in SEAK than MB for droplet and teepee calls (Table 3, Fig. 4), indicating that calls from SEAK were generally higher pitched than calls from MB.

Figure 3 Boxplots of PC1 values (indicative of entropy, bandwidth, and upper frequency components) between call types and ocean basins.

Calls recorded in the Atlantic Ocean are indicated by coral, and the Pacific ocean by teal. Call types: (A) droplet; (B) growl; (C) swop; (D) teepee; (E) whup. Boxplots illustrate median, first, and third quartile PC1 values; dots indicate outliers.

Figure 4 Boxplot of PC2 values (indicative of lower frequency, start frequency, and peak frequency components) between call types and ocean basins.

Calls from the Atlantic are indicated by coral, calls from the Pacific are indicated by teal. Call types: (A) droplet; (B) growl; (C) swop; (D) teepee; (E) whup. Boxplots illustrate median, first, and third quartile PC2 values; dots indicate outliers.

Discussion

This is the first study to describe call types shared by allopatric humpback whale populations. Evidence that temporally stable call types are shared between Southeast Alaska (SEAK) and Massachusetts Bay (MB) humpback whale populations supports the hypothesis that a portion of the call repertoire may be fixed in this species.

In SEAK there are six call types that are stable over generational time (Fournet, 2018) that have average bandwidths between 10 and 1,000 Hz: droplets, growls, swops, teepees, whups, and feeding calls. Misclassification was low for all call types, except for whups, which were commonly classified as growls. Misclassification of these call types is unsurprising, as the only distinguishing acoustic feature between growls and whups is a terminal upsweep, which attenuates with distance and is not adequately encompassed by traditional acoustic parameters (Fig. 2) (Fournet, Szabo & Mellinger, 2015). The humpback whale call repertoire has been described as an acoustic continuum, where graded signals are common (Rekdahl et al., 2013; Fournet, Szabo & Mellinger, 2015). The delineation between growls and whups is not discrete, and it is currently unknown whether whups and growls are functionally interchangeable. Methods for either classifying graded signals or more broadly aggregating them according to their functional roles merits future investigation.

Within-call variation, related to individual anatomy, behavioral or environmental context can be found within most if not all vertebrate vocalizers, and does not contradict placement into call classes or types (Ford, 1991; Tyack & Sayigh, 1997; Deecke, Ford & Spong, 2000; Tibbets & Dale, 2007; Rekdahl et al., 2013). In this study, despite otherwise high classification agreement, there were some differences in call type parameters between populations. The increased PC1 values found in SEAK versus MB may be recording artifacts. The ambient sound conditions in SEAK are significantly different than MB (Kipple & Gabriele, 2003; Hatch et al., 2008; Haver et al., 2018). Recordings from Frederick Sound were made in the absence of vessel noise, and recordings made in GBNPP were made in the presence of limited vessel traffic. By contrast, the hydrophones in MB were located within a shipping lane that services Boston Harbor, which is among the busiest harbors on the North American east coast. For this reason, vessel noise was recorded simultaneously with almost all calls recorded in MB (Fig. 2). Overlapping ambient sounds—including vessel noise, which is common throughout the 10–1,000 Hz band (Wenz, 1962)—may have masked fine-scale acoustic features, resulting in decreased aggregate entropy measurements in MB calls. Similarly, vessel noise in MB may have masked upper-frequency portions of calls, which contain less energy and attenuate faster and are thus more easily obscured by overlapping ambient sound. Systematic differences in frequency between droplets and teepees in SEAK vs. MB (Table 3) may be related to factors such as motivational state (Rehn et al., 2011; Dunlop, 2017), body size (May-Collado, Agnarsson & Wartzok, 2007), and/or ambient noise (Parks, Clark & Tyack, 2007; Di’Iorio & Clark, 2010; Parks et al., 2016), but a dedicated research effort that includes direct observation and identification of individuals would be required to address this question.

With one exception, call types of interest from SEAK were also found in MB. The notable exception was the SEAK feeding call. Feeding calls are highly stereotyped, tonal calls, with a fundamental frequency of ∼500 Hz that occur when humpback whales in Southeast Alaska forage on Pacific herring (Clupea palisii) (D’Vincent, Nilson, & Hanna, 1985; Sharpe, 2001; Fournet et al., 2018). Herring are a primary food source for humpback whales in Southeast Alaska (Krieger & Wing, 1984; D’Vincent, Nilson, & Hanna, 1985; Dolphin, 1988), whereas in MB humpback whales feed primarily on sand lance (Ammodytes spp.), a calorie-dense prey species that burrows in the sandy substrate (Overholtz & Nicolas, 1979; Hain et al., 1995; Friedlaender et al., 2009). The absence of feeding calls in MB may be attributed to their focus on forage species other than herring.

Droplets, growls, swops, teepees, and whups were present in the call repertoire of both humpback whale populations. Evidence of the same calls in allopatric populations supports the hypothesis that a portion of the humpback whale call repertoire is innate. Many non-passerine bird species such as doves (Streptopelia sp.) produce highly stereotyped calls instinctively (Lade & Thorpe, 1964), and as a result allopatric dove populations of the same species, even those separated by great distances, show no significant difference in call types (De Kort, Den Hartog & Ten Cate, 2002). Ornate chorus frogs (Microhyla fissipes) produce advertisement calls independently of vocal learning that are aurally indistinguishable between geographic regions, and that vary only minutely with genetic distance (Lee et al., 2016). Genetic predetermination of calls is common across taxa, including zebra finches (Taeniopygia guttata; Forstmeier et al., 2009), fur seals (Antarctic, Arctocephalus gazella, subantarctic, A. tropicalis, and New Zealand, A. forsteri; Page, Goldsworthy & Hindell, 2001), and Spheniscus penguins (Thumser & Ficken, 1998). Call type stereotypy in these species is generally multi-generational and geographically widespread. In humpback whales, identifying call types that are multi-generational, and persist in geographically and genetically discrete populations provides strong evidence that these call types are innate.

If the call types described in this study are innate to humpback whale worldwide, as we hypothesize, then it should be possible to build an automated acoustic detector that could be run on datasets from across ocean basins and years to confirm the presence of humpback whales at previously unknown regions or times. The ability to confidently credit particular vocalizations to humpback whales in the absence of visual confirmation allows for broader spatial and temporal monitoring with significantly lower effort and cost (see also Stimpert et al., 2011).

For calls to be conserved within the call repertoire of genetically and geographically discrete populations is an indication that they play an important role in humpback whale life history by increasing individual fitness in some capacity. It has been proposed that in Southeast Alaska the whup call serves a contact function (Wild & Gabriele, 2014), and the analogous “wop” call of east Australia may facilitate communication between cows and calves (Dunlop, Cato & Noad, 2008). There is also evidence that droplets, swops, and teepees are used for close range communication on foraging grounds (Fournet, 2014), and similar pulsed calls may facilitate affiliation or disaffiliation in groups during migration (Dunlop, Cato & Noad, 2008). These broad contextual descriptions, suggest that these calls serve a vital function or functions. The fixed nature of calls stands in marked contrast to humpback whale song, which is geographically discrete, changes rapidly, and is culturally transmitted rather than innate (Payne & Payne, 1985; Noad et al., 2000; Cerchio, Jacobsen & Norris, 2001). Thus, it seems that the humpback whale vocal repertoire is composed of both fixed and adaptable calls. Dedicated research pairing the call types described in this study with behaviors and social context will further the understanding the role of calls in the acoustic ecology of humpback whales.

Conclusions

This study demonstrates that some humpback whale call types are shared between geographically discrete northern latitude foraging grounds. This feature lend support to the hypothesis that some calls may be innate, and in strong contrast to song, are not culturally transmitted. Natural next steps include a global comparison of call repertoires between allopatric populations and across the migratory range, with particular attention paid to change or stability at various temporal and geographic scales.

Supplemental Information

Supplemental Information 1 Spreadsheet including acoustic parameters and call type assignments for each recording location

Click here for additional data file.

The authors wish to acknowledge Dr. Roger Payne for the use of the recordings from Southeast Alaska in 1976. We thank David Culp for data processing support, and Katherine Indeck for statistical support. We also wish to thank the National Park Service for its long term commitment to acoustic monitoring in Glacier Bay National Park, and to the Cornell Bioacoustics Research Program for use of the data from Massachusetts Bay. This is PMEL contribution number 4784.

Additional Information and Declarations

Competing Interests

Author Contributions

Animal Ethics

Data Availability

The authors declare there are no competing interests.

Michelle E.H. Fournet conceived and designed the experiments, performed the experiments, analyzed the data, prepared figures and/or tables, authored or reviewed drafts of the paper, approved the final draft.

Lauren Jacobsen analyzed the data, approved the final draft.

Christine M. Gabriele contributed reagents/materials/analysis tools, authored or reviewed drafts of the paper, approved the final draft.

David K. Mellinger and Holger Klinck contributed reagents/materials/analysis tools, approved the final draft.

The following information was supplied relating to ethical approvals (i.e., approving body and any reference numbers):

This work was done passively without interaction with any animal subjects and did not require a animal care permit.

The following information was supplied regarding data availability:

The raw data are provided in Supplemental File.

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
