# Peer review of "More of the same: allopatric humpback whale populations share acoustic repertoire"

_PeerJ, doi:10.7717/peerj.5365_

## Round 0.1 · original submission · Major Revisions

Dear authors,

Thank you for submitting your manuscript to PeerJ. Although it looks promising, a number of issues were raised by the two external referees regarding the technical soundness of your paper.Please address to each point carefully in a revision and the rebuttal letter. I fully agree with them and in particular also to ref 2 pointing to major inconsistencies in the statistical soundness of your paper...After submission of your revised MS, it will be sent out again to external referees before a decision can be made.

We are looking forward to your revision.

Best wishes
Elke Zimmermann

·

Basic reporting

Very well written, with only a few general comments:

Line 56: Perhaps define pinniped and cetacean upon initial use as this journal is aimed at a general audience.

Line 75: Remove “and”, and start a new sentence with “Further”.

Lines 120-127: This is slightly confusing. Perhaps note that Frederick Sound is in SEAK and is where the 1976 recordings were made and not just have that information in the figure alone.

Line 165: For the reader that is not versed in acoustics, it might be helpful to provide the call types that fit into the category of harmonic sounds vs pulsed sounds here as well as Table 3. I find it helpful to read short descriptions of calls as well as looking at spectrograms. I understand there might be word limit or other constraints, but it is nice to know additional information about these calls if it is available (always given in bouts? ever variable?)

Line 192: Remove the word “with”.

Line 221: Should be “pitched” not “pitch”.

Line 258: See also Parks et al. 2016 "Noise impacts on social sound production by foraging humpback whales" for a more recent citation.

Line 308: Insert the word “the” between “understanding” and “role”.

Figure 2: The x axis label (Time) is not centered, and the x axes on each separate panel are a bit confusing - two separate calls should have separate time tick marks. The headers "Pacific" and "Atlantic" are not centered above their spectrograms. It seems "teepee" should be plural if "swops" is. In the actual uploaded figure file, the axes are not present.

Figure 3: It would be nice to have significance noted on the figure with asterisks, perhaps next to the call type name. Also, again, if "swops" is plural, "teepees" should be as well.

Figure 4: Same regarding “teepees” being plural.

Table 1: Some of the rows in this column don't line up well with the rows in the last three columns - they are just slightly off enough to disrupt the flow while reading. This is likely due to the words being centered in the cell. Consider adding grid lines to facilitate reading.The words “unknown” and “unavailable” should be capitalized for consistency with the rest of the table.

Table 2: Labelled incorrectly in the caption (labelled as Table 1 again). Also, PC2 should be defined as “rotated” principal component, rather than “rotate” principal component.

Table 3: The broad call type classification (harmonic vs pulsed) is not really discussed elsewhere in the text. I like to see broad categories, as often call types are often graded and not always as well split with CART and RF as they are here. I would suggest perhaps adding this briefly to the main text.

Table 4: Capitalize “Regression” and “Tree” in the caption.

Experimental design

Line 169: I appreciate citations in methods for brevity, however this reference is for your thesis which is not yet accessible. Is it possible to include either an additional reference to work you have already published, or to provide some information on the quantitative classification methods used here?

Line 171: Could you please provide the package information? Additionally, it would be helpful to provide a brief explanation of what each of these analyses offers statistically, and more specifically, why you chose to use each.

Line 188: Why did you use an older version of R? Does this mean that the packages used were also older versions? Packages can sometimes be updated to reflect advances in statistical methods.

Validity of the findings

No comment.

Reviewer 2 ·

Basic reporting

no comment

Experimental design

no comments

Validity of the findings

no comments

Comments for the author

The authors studied two allopatric humpback whale populations (in the North Pacific Ocean and North Atlantic Ocean) and showed that both populations shared five call types, which provides evidence that these call types may be innate. The spectrograms in Fig. 2 showed that both populations used call types with a very similar structure. But their analytical approaches are more confusing than helpful (see below).
1) General question (L50-64): “What is less common among mammalian vocal learners, however, is the coupling of stable sound types, which may be innate, with a dynamically changing repertoire of sound types whose variation appears to be culturally driven.”
That is not true. Similar like birds, mammalian vocal learner have a rich repertoire of innate calls. This is true for mice (vocal learners?), pinnipeds, bats or elephants. It is less clear for some cetacean species because most studies were done to emphasize their vocal flexibility. Insofar this is an interesting study but it is wrong to say that it would be surprising that mammalian vocal learner have innate calls.
2) It remains unclear how the calls from the North Pacific recorded in 1976 differ from the 2007 recordings and whether this has an influence on the results.
3) L. 147-49: “Frequency parameters were log-transformed to account for the mammalian perception of pitch, which is approximately logarithmic rather than …”.
If such a transformation is necessary at all, the authors should use a transformation related to the pitch perception of humpback whales. Mammalian pitch perception could be very different.
4) L.170-73: The validity of forest analysis remained unclear, especially in combination with an additional observer validation. I think it would be more objective to use just the assignment result of this analysis, or any other procedure which are able to compare the similarity of two samples (population).
5) Table 2:
Aggregate entropy is under such conditions not a measure of total disorder in the call, because this measure is mainly influenced by recording conditions, like distance of caller, different equipment, etc. I think it makes no sense to use a variable which is mainly influenced by the unknown distance of the signaler.
6) Results in Table 3:
The authors have no knowledge about the number of subjects. Therefore the use of Kruskal-Wallis test or similar tests is incorrect. There are possibilities to deal with such data sets (e.g. Vester et al. 2016 Physical Review). In addition, it remains unclear what the authors like to show with these tests. There are high significant p-values. Should this mean that both populations differ in their call types?
6) L-197-98: “PCA output indicated that the use of two principal components was adequate to encompass the variability of the data (χ2= 683.29, p < 0.00001).”
It remains unclear what this result should tell. And why did the authors use in addition to the two PCs also the other acoustic variables? They just showed that the two principle components were adequate to encompass all the variability.
7) L. 210-12: “The variables most important for splitting decisions in the random forest analysis were bout, end frequency, duration, entropy, lower frequency, PC1, PC2, and frequency trend.”
Again, why using both PCs the other acoustic variables. What means most important in case nearly all acoustic variables a most important?
8) L. 213-14:
What is the value of this mixture between analytical assignment (CART) and observer validated assignment. The call type categories were established by observer criteria. Therefore one should be able to achieve an agreement between different observers, especially by omitting critical calls.

---

## Round 0.2 · Major Revisions

Although your revised paper has improved, still some issues were raised by the external referee regarding the technical soundness of your paper. The referee further noted that you do not have responded to these issues in your revision. Thus, please address to each point carefully in a re-revision and the rebuttal letter. I fully agree with the referee pointing to major inconsistencies in the methodology and statistical soundness of your paper..Please note: this will be your last possibility to improve your paper. After submission of your re-revised MS, I will make a final decision.

Best wishes
Elke Zimmermann

Reviewer 2 ·

Basic reporting

Resubmission.

Experimental design

Resubmission.

Validity of the findings

Resubmission.

Comments for the author

The authors gave answers to all my comments, but ignored my suggestions. Therefore I cannot really say that the manuscript has improved. But as I mentioned in my first review, the conclusions are supported by the results. Humpback whales of the North Pacific Ocean share call types with conspecifics in the North Atlantic Ocean.
But I do not agree with following explanations:
1) L 154-57: “...; all recordings were made with 16 bit resolution; once resampled recording were technically comparable in all aspects.”
Just use the same amplitude accuracy to resample the recordings does not mean to have technically comparable recordings. The recordings remain different in relation to recording equipment and conditions.
2) L 234-47; Wilcoxon rank-sum test:
I do not see a good reason why the authors used this test. They do not know their real sample size; therefore it is not possible to use this test.

3) L 259-60:
The classical description of the quality a factor analysis is KMO, Bartlett's test, and the explained variance of the factors. A p-value of 0.00001 is not very informative. It would be difficult to produce a non-significant factor solutions.

---

## Round 0.3 · Minor Revisions

Thank you for considering the critical comments and suggestions of the former referee in the re-revised MS.

I have only minor issues which have to be considered before the MS can be fully accepted:

1. Line 57: delete "mouse" and focus to sea-living mammals. Findings on mice are discussed highly controversial since there are more paper supporting an innate vocal repertoire. Besides, bats and elephants would provide much better examples for vocal production learning.
2. Line 197: Control if you really mean "bootsrapping", instead of "bootstrapping" or if there is a typing error.
3. Line 300-311: provides not really convincing arguments to explain a potential geographical distinctiveness in call types between the two populations. According to your arguments, comparable divergences should appear across all call types within a population. Table 3 is important and really necessary since it contains "real" acoustic data. Looking at it suggests geogaphical distinctiveness in at least one call type (Teepee), but not others. This does not confirm that all call types, besides song, are similar between the two populations, as you suggest in title and paper. Please consider this part of the Discussion again and provide a more convincing explanation.
4, Control Tables. Table 1 appears twice in my PDF.
5. Table 2 acoustic parameters.... As one referee pointed out, I also think that "Aggregate Entropy" is not a good parameter in your set up since it depends strongly on the sound environment for your calls which is highly variable, as you describe. Please explain in the Methods why you think it is necessary to keep this unreliable parameter in your analysis.
Please exclude the PC1 and PC2 components from the Table. They merit a separate section in your Methods, in which you explain, what you have done here and why and how you produced with them the graphs with median (?), quartiles (?), outliers (?) in Figure 3 and 4.
6. Please tone down the first sentence in the conclusion. You have not demonstrated in this paper that call types between the two studied populations are temporally stable or do not differ statistically in their acoustic parameters.This would have required a totally different set-up.

Please also carefully address the guidelines for this journal.

---

## Round 0.4 · accepted · Accept

Thank you for revising the MS which I will accept in its current form. It will be further processed by the technical staff of PeerJ.

Best wishes
Elke Zimmermann

#